# Anthrax outbreak investigation in Tengwe, Mashonaland West Province, Zimbabwe, 2022

**Theresa Hamutyinei Dhliwayo**[1☯], **Prosper Chonzi**[2☯], **Collen Madembo**[2☯], **Tsitsi Patience Juru**[1☯], **Addmore Chadambuka**[1☯]*, **Notion Tafara Gombe**[1,3☯], **Silvester Chikerema**[4☯], **Gerald Shambira**[1☯], **Chukwuma David Umeokonkwo**[3☯¤], **Mufuta Tshimanga**[1☯]

**1** Department of Primary Health Care Sciences, Family Medicine, Global and Public Health Unit, University of Zimbabwe, Harare, Zimbabwe, **2** Harare City Health Department, Harare, Zimbabwe, **3** African Field Epidemiology Network, Harare, Zimbabwe, **4** Department of Clinical Veterinary Studies, University of Zimbabwe, Harare, Zimbabwe

☯ These authors contributed equally to this work.
¤ Current address: African Field Epidemiology Network, Monrovia, Liberia
* achadambuka1@yahoo.co.uk

## Abstract

### Background

Anthrax continues to be a disease of public health concern in Zimbabwe. Between December 2021 and February 2022, Tengwe reported 36 cases of human anthrax. Gastrointestinal anthrax has the potential to cause serious outbreaks leading to loss of human life. We investigated the outbreak, identified the risk factors using one health approach to inform outbreak control.

### Materials and methods

We conducted descriptive analysis of the outbreak and a 1:2 unmatched case control study to identify risk factors for anthrax. A case was any Tengwe resident who developed an ulcer and/or abdominal symptoms and epidemiologically linked to a confirmed environmental exposure. Validated, structured interviewer-administered questionnaires were used to collect data from the cases and neighbourhood controls. Soil and dried meat samples were collected for laboratory investigations. District preparedness and response was assessed using a checklist. Data was analysed using Epi Info version 7.2.5. The odds of exposure were calculated for each risk factor examined. Multivariable logistic regression analysis was performed to identify the independent factors associated with contracting anthrax.

### Results

Through active case finding we identified 36 cases, 31 were interviewed. Twenty-one (67.7%) were males. The median age was 33 years (Inter quartile range: 22–54). Nineteen (61.2%) cases presented with abdominal symptoms with zero deaths reported. The independent risk factor for contracting anthrax was eating under-cooked meat (aOR = 12.2, 95% CI: 1.41–105.74, p = 0.023). All samples collected tested positive for anthrax. No livestock vaccinations or zoonotic meetings were being conducted prior to the outbreak. Notification

**Data Availability Statement:** Dataset has been uploaded to Figshare and the DOI is https://doi.org/10.6084/m9.figshare.21610206.v1.

**Funding:** The author(s) received no specific funding for this work.

**Competing interests:** The authors have declared that no competing interests exist.

of the outbreak was done 11 days after index case presentation however one health response was instituted within 24 hours.

## Conclusion

The anthrax was confirmed in Tengwe. Consumption of under-cooked meat was associated with gastrointestinal anthrax. The timely one health response resulted in excellent outcomes. Using one health approach in managing zoonotic threats is encouraged.

## Introduction

Anthrax, a primarily disease of the herbivores, is a widespread zoonosis caused by a Gram-positive rod-shaped bacterium called *Bacillus Anthracis* [1]. The bacteria can enter the human body through ingestion, inhalation or skin (cutaneous) where the latter is the most common form [2].

Human anthrax cases are still common in Africa and Asia despite the decrease in global incidence [3]. The largest outbreak of human anthrax in the world occurred in Zimbabwe between 1978 and 1980 where 10738 cases and 182 deaths were recorded. Since this outbreak, anthrax has become endemic in Zimbabwe with sporadic outbreaks of the disease [4].

The incubation period for cutaneous and gastrointestinal anthrax ranges from 12 hours to 12 days [5]. Common symptoms include fever, an ulcer (eschar), abdominal pain, vomiting and diarrhea. In severe cases, there may be haemorrhage (commonly manifesting as bloody diarrhoea), intestinal obstruction or perforation, sepsis, or shock which may result in deaths [6]. Without treatment, the case fatality for the cutaneous and gastrointestinal form is approximately 20% and 40% respectively [7].

The prevention and control of anthrax is an inter-sectoral activity. Control measures are aimed at breaking the cycle of transmission, mainly based on its prevention in livestock through mass vaccination and proper disposal of infected carcasses. Vaccination should be done annually in high risk areas [8]. Compulsory dipping in the communal lands has led to the reduction of anthrax cases in Zimbabwe [9].

The Veterinary department for the district first reported the death of a cow with suspected anthrax on 16 December 2021 and by 10 January 17 had died. A total of 36 people developed symptoms suggestive of anthrax. The index case was as a 24-year-old male who presented on the 27th of December 2021 at the government district hospital with bloody diarrhoea and ulcers on the 4th and 5th right fingers. He was sent home without the diagnosis of anthrax being made. He then presented at a private clinic where diagnosis of anthrax and treatment was done, however notification was not done. The second case presented on 7 January 2022 at the government district hospital where a diagnosis and notification of the anthrax was made. By 9 January, 13 cases had been reported and by 10 January, 31 cases. Both the provincial and district rapid response teams went to the area to actively identify other cases as well as contain the spread of the disease. The One health approach was in place in response to this outbreak as the veterinary department, human health and environment constituted the team that investigated the outbreak.

Investigation of this anthrax outbreak was relevant to provide immediate evidence-based decisions to inform current and future control measures by first assessing the level of preparedness of the district. To our knowledge, except for one reported intestinal anthrax during 1979 major outbreak there are no other published reports of gastrointestinal anthrax in

Zimbabwe, therefore we sought out to investigate and determine the factors associated with this outbreak [4].

## Materials and methods

### Study setting

Mashonaland West is the second largest province of the ten in Zimbabwe and is situated in the North Central region [10]. The province has seven districts with Hurungwe being the largest. Tengwe area is situated approximately 40 km south of Hurungwe District [11].

The area was previously occupied by large scale commercial farmers where cattle ranching and tobacco farming were practiced. These farms have been downgraded to small scale, and for the past four years, no routine annual vaccinations by the Veterinary Department were being done. The area receives high rainfall with an average of 700–800mm per year [12]. The population is approximately 3000 people according to the 2020 districts' estimates. No cases of human anthrax had been reported in this area before.

### Study design

A descriptive study was carried out first to describe the outbreak in person, place and time. A line list was developed from the active case finding. An analytic study to test the hypothesis on whether religions with restrictive beliefs was associated with contracting anthrax was conducted. This was a 1:2 unmatched case control study where the outcome of interest was those who had anthrax symptoms.

### Study population

Our study population was residents of Tengwe, Hurungwe District. The district managers (District Medical Officer (DMO), District Environmental Health Officer (DEHO), Senior Community Nurse, District Veterinary Officer) were interviewed as key informants.

Case: Any individual who resided/visited Tengwe between 16 December 2021 to 13 February 2022 and developed clinically compatible sign and symptoms of anthrax (skin lesion/s evolving from a papular through a vesicular stage, to a depressed black eschar invariably accompanied by oedema that may be mild to extensive and/or abdominal distress characterized by nausea, vomiting, diarrhoea anorexia and followed by fever) and epidemiologically linked to a confirmed environmental exposure [13].

Control: Any person who resided/visited Tengwe, for the same period, who could be a household member or neighbour and did not develop any of the signs and symptoms mentioned above, that are consistent with infection by *Bacillus anthracis*.

Inclusion criteria: People of all age groups who were residents/visitors of Tengwe between 16 December 2021 to 13 February 2022 who developed clinically compatible symptoms of anthrax, were epidemiologically linked to a confirmed environmental exposure and consented to be part of the study.

Exclusion criteria: People of all age groups who were residents/visitors of Tengwe, between 16 December 2021 and 13 February 2022 but were not able to give written informed consent, were ill and not able to sustain the interview and those who were not available on two consecutive visits.

### Sampling and sample size

All cases obtained from the line list produced from the health centre based medical records were included in the study. At each household where a case was included, two controls were

randomly chosen through picking a card from a hat. If none or only one control was found the next house was chosen for the control where the same process was repeated.

## Data collection

An interviewer administered questionnaire was used to collect information on the demographic characteristics, knowledge and practices and factors associated with contracting anthrax from the study participants. A checklist was used to evaluate quality of outbreak detection, investigation, and response. The checklist assesses the time periods for notification, investigation, reporting, availability of medicines against the set standards. Members of Hurungwe District Health Executive (DHE) were interviewed as key informants.

Records from the district veterinary department regarding the district livestock population and deaths of cows, sheep and goats were reviewed. The records on livestock vaccination campaigns conducted by the district were also reviewed.

## Environmental sampling and laboratory detection of *B. anthracis*

Soil samples were collected from carcass burial sites. Dried meat samples were collected from the door-to-door inspections. Isolation of *B. anthracis* from soil and dried meat was done according to the method of Fasanella *et al*, 2013 with some modification where 7.5g of soil/dried meat was mixed with 22.5ml sterile deionised water and 0.5% Tween 20 [14]. After vortexing, the supernatant was incubated at 65˚C for 20mins. 1ml/plate of the supernatant was incubated in B anthracis selective (PLET) media at 37˚C for 36-48hrs. Colonies that appeared small, white, round, and domed were subcultured in sheep blood agar for overnight incubation. Grey-white irregular colonies that were Gram positive, non- haemolytic, catalase positive, oxidase negative, non-motile and had a positive Mcfadyean reaction were regarded as B anthracis. Human stool samples taken were not tested due to unavailability of reagents.

## Data analysis

Epi Info7 (2021 $^{TM}$) statistical software was used to capture and analyse the data. Frequencies for all the variables were used to check for missing variables and the questionnaires were used to correct any mistakes in the consistency of data entry. Univariate analysis, that is the calculation of frequencies, means and proportions was done. On bivariate analysis, variables were measured against the outcome of interest which was being ill with anthrax. Religion was further divided into two groups, restrictive beliefs, and non-restrictive beliefs, where a restrictive belief was one which forbad eating of animals that died on their own and the later permits. A total of 20 questions were used to assess knowledge. The overall knowledge was categorized as good for a score of 50% and above, poor for below 50%. Odds ratios were calculated from the bivariate analysis and their 95% confidence intervals were calculated. To control for confounders as well as report on different effects seen for effect modifiers, stratified analysis was done for sex and age. To identify independent factors, multivariate analysis using a stepwise backwards logistic regression model was performed. All variables with a p-value $\leq 0.25$ were included in the logistic regression model. All variables with a p-value $<0.05$ were statistically significant. Data was presented in the form of tables, maps, and graphs.

## Ethical considerations

Written informed consent to participate in the anthrax outbreak investigation was obtained from all participants. Consent for minors was sought from the parents or guardians. Participation was voluntary with respect of choice on whether to participate or not being maintained.

Confidentiality and security of collected data was assured and maintained, no information leading to patient identification was collected. Purpose, risks/discomforts, and benefits of participating in the study were explained to all participants.

**Permission to proceed.** Permission to carry out the program evaluation was obtained from Harare City Health Director, Health Studies Office, Mashonaland West Provincial Medical Directorate, Hurungwe DMO and Community Leaders.

## Results

### Descriptive epidemiology

We described the outbreak in person place and time. The results are shown below:

**Distribution by person and place.** A total of 36 cases were identified by both active and passive surveillance. Thirty-one cases were interviewed as one person was not willing to participate while four were not available at their homesteads after two consecutive visits. There were more males than females, 21(67.7%) vs 10 (33.3%) respectively. The median age was 33years ($Q_1$ = 22 years; $Q_3$ = 54 years). The median number of people in the households was 6 ($Q_1$ = 5, $Q_3$ = 8) (Table 1). All cases had been residing in Peterlands farm, Tengwe during the outbreak period.

**Distribution by time.** The index case started having symptoms on the 23rd of December 2021 and presented at the hospital on 27 December 2021. Notification to the DHE and Veterinary department was done on 7 January 2022 after presentation of the second case. Intensive case finding and health education started on 8 January 2022. The veterinary department started cattle vaccination on 10 January 2022. Cases continued to be reported until the 20th of January 2022 and the outbreak was declared over on the 13th of February 2022 (Fig 1).

**Clinical presentation.** Bloody diarrhoea 19/31 (61.3%) was the most common symptom identified, with other symptoms identified being, cutaneous eschar 18/31 (58.1%), cellulitis 12/31 (38.7%), abdominal pain 9/31(29%), generalised body weakness 5/31(16%), headache 3/31 (10%) and fever (Fig 2). For the cutaneous form, 13/15 (87%) lesions were on the forearms and hands, one on the face and the other one behind the left ear.

**Knowledge on anthrax.** All the participants had heard of the disease anthrax. The most common identified symptom was a skin lesion (90% for both cases and controls). More cases compared to controls were able to identify the other symptoms of anthrax.

**Evaluation of the outbreak preparedness, detection and response.** An eight-member team which consisted of 4 members from the human health department, 3 from the veterinary and one from the environment went into the field to investigate this outbreak. We found out that the district hospital had adequate stocks of the recommended antibiotics for the empiric treatment of anthrax and treatment guidelines were available in the form of standard treatment guidelines (Essential Drugs List of Zimbabwe, 7th Edition) [15]. The laboratory did not have the capacity to do microbial cultures. The District Epidemic Management Committee and District zoonotic committee did not hold meetings and there were no minutes or plans to review.

A total of 17 cattle deaths were reported however no specimens from these were collected by the district veterinary department for confirmation of anthrax. Only one carcass of a suspected animal was disposed under the supervision of a veterinary officer. A search of the suspected infected animal products was done in homes where some dried meat was found and disposed by burying with the help of area veterinary officers and environmental health workers. A quarantine order for closure of all abattoirs and butcheries was issued on the 10th of January 2022 by the Veterinary department and was lifted after 28 days.

Soil samples were taken by the Veterinary department for analysis. Soda of chloride was not available for disinfection of carcasses and infected soil. Combined health sessions to the community were also done, where all the three departments were available. We gathered

**Table 1. Socio-demographic characteristics of human anthrax cases and controls, Tengwe, Hurungwe District, 2022.**

| Variable | Case n = 31 (%) | Control n = 62(%) |
|---|---|---|
| Sex | | |
| Female | 10 (33.3) | 29 (46.8) |
| Male | 21 (67.7) | 33 (53.2) |
| Age (years) | | |
| 11–20 | 16 (19.4) | 24 (38.7) |
| 30 | 7 (22.6) | 15 (24.2) |
| 31–40 | 10 (32.3) | 6 (9.7) |
| 41–50 | 6 (19.4) | 6 (9.7) |
| > 50 | 2 (6.5) | 11 (17.7) |
| Median age (*IQR) | 33 (22–54) | 23 (20–76) |
| Level of education | | |
| Primary | 9 (29.0%) | 1 (1.6) |
| Secondary | 21 (67.8%) | 12(19.4) |
| Tertiary | 1 (3.2%) | 49 (79.0) |
| Religion | | |
| Apostolic | 7 (22.6%) | 26 (41.9) |
| Catholic | 2 (6.5%) | 1 (1.6) |
| Seven Day Adventist | 4 (12.9) | 8 (12.9) |
| Pentecostal | 5 (16.1%) | 11 (17.7) |
| Mainline | 5 (16.1) | 4 (6.6) |
| Other | 2 (6.5%) | 1 (1.6) |
| None | 6 (19.4%) | 11 (17.7) |
| Marital Status | | |
| Single | 11 (35.5%) | 18 (29.0) |
| Married | 18 (58.0%) | 39 (62.9) |
| Divorced | 2 (6.5%) | 2 (3.2) |
| Widowed | | 3 (4.9) |
| Household size | | |
| ≤ 3 | 2 (6.5) | 1 (1.6) |
| 4–6 | 14 (45.1) | 31 (50.0) |
| 7–9 | 11 (35.5) | 19 (30.6) |
| ≥ 10 | 4 (12.9) | 11 (17.8) |
| Median household size (*IQR) | 6 (5–8) | 6 (5–9) |

*IQR- Inter Quartile Range

information from the community that a herd of buffalo was seen in the area prior to the death of cows. Communication was done with the wild parks department who reported no suspected anthrax cases during this period. No cattle vaccinations were being done in the district over the past 4 years. During the outbreak, the district vaccinated 46 000 cattle (98% coverage) in the affected and surrounding areas.

**Factors associated with contracting anthrax.** Being female (OR 0.54, p = 0.183) and belonging to a restrictive religion (OR 0.86, p = 0.820 and were found to be protective factors. For the hypothesis of the study: Belonging to a restrictive religion is not associated with contracting anthrax, the p- value was > 0.05 therefore we failed to reject the null hypothesis. Significant factors associated with contracting anthrax included skinning a carcass without protection (OR 3.01, p = 0.021) and eating undercooked meat (OR 16.5, p < 0.001) (Table 2).

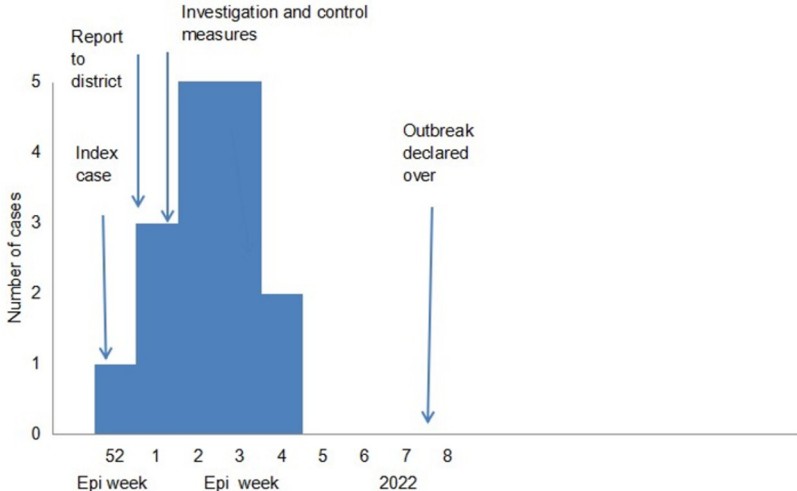

**Fig 1. Epi-curve of the human anthrax cases in Tengwe, Hurungwe District 2022.**

**Stratified analysis.** On performing stratified analysis, we found that the risk of contracting anthrax from cutting and cooking infected meat was higher in females (OR 8.4, p = 0.032) as compared to males (OR 1.71, p = 0.045). Gender was found to be a confounder with the adjusted OR (MH) of 3.27.

**Independent risk factors associated with contracting anthrax.** We included the following factors in our logistic regression analysis: eating undercooked meat, skinning a carcass, and cooking infected meat and preparing hides. The independent risk factor associated with contracting anthrax was eating undercooked meat (a OR = 12.2, 95% CI: 1.41–105.74, p value—0.023).

## Discussion

Our investigation documented a cutaneous and gastrointestinal anthrax outbreak with no fatalities. The outbreak was associated with consuming infected meat in a new resettlement

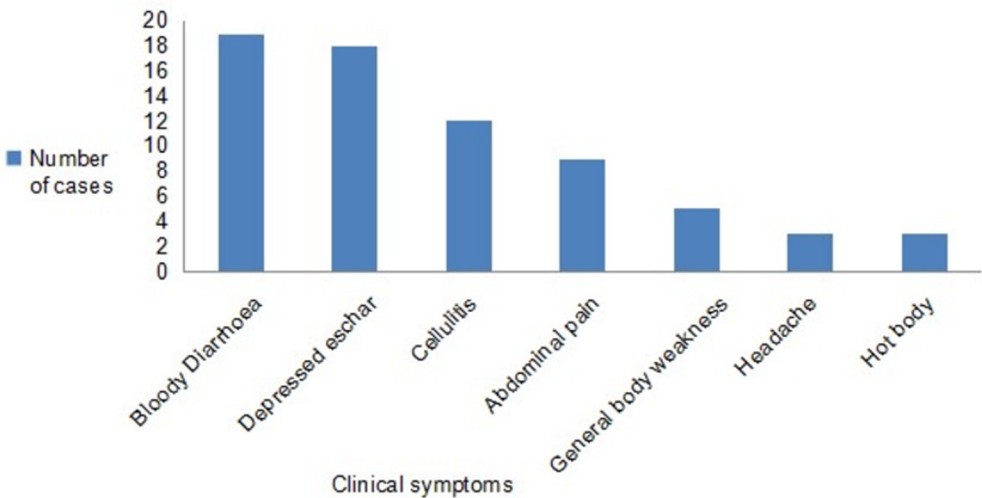

**Fig 2. Clinical presentation of human anthrax Tengwe, Hurungwe District, 2022 (n = 31).**

**Table 2. Factors associated with contracting anthrax, Tengwe, Hurungwe District, 2022.**

| Variable | Cases | Controls | OR (95% CI) | p-value |
|---|---|---|---|---|
| Sex | | | | |
| Female | 10 | 29 | 0.54 (0.2–1.34) | 0.183 |
| Male | 21 | 33 | Ref | |
| Level of education | | | | |
| Primary or less | 9 | 13 | 1.54 (0.5–4.14) | 0.442 |
| Secondary or above | 22 | 49 | Ref | |
| Religious beliefs on consumption of meat | | | | |
| 'Restrictive' | 27 | 55 | 0.86 (0.23–3.19) | 0.820 |
| Non 'Restrictive' | 4 | 7 | Ref | |
| Household size | | | | |
| < = 6 | 16 | 32 | 1.00 (0.42–2.37) | 1.00 |
| >6 | 15 | 30 | Ref | |
| Eating undercooked meat | | | | |
| Yes | 30 | 40 | 16.5 (2.11–129.36) | <0.001 |
| No | 1 | 22 | Ref | |
| Skinning of carcass without protection | | | | |
| Yes | 13 | 12 | 3.01 (1.16–7.80) | 0.021 |
| No | 18 | 51 | Ref | |
| Preparing meat | | | | |
| Yes | 19 | 23 | 2.68 (1.11–5.52) | 0.027 |
| No | 12 | 39 | Ref | |
| History of a dead beast in the homestead | | | | |
| Yes | 19 | 23 | 2.68 (1.11–6.52) | 0.027 |
| No | 13 | 39 | Ref | |
| Assisting in carcass disposal | | | | |
| Yes | 3 | 3 | 2.11 (0.40–11.1) | 0.371 |
| No | 28 | 59 | Ref | |
| Overall knowledge on anthrax | | | | |
| Good | 19 | 23 | 2.48 (1.03–5.94) | 0.040 |
| Poor | 13 | 39 | Ref | |

[a]Restrictive- does not permit consumption of meat from animals that died on their own, animals killed due to unknown causes or butchered after an unobserved death

area in Hurungwe District, Zimbabwe. Notification of the outbreak was late however the One health response was instituted within 24 hours.

The outbreak occurred in an area where annual vaccinations by the veterinary department had stopped four years prior. It meant that the cattle herd were not protected from possible outbreaks. This is consistent with a study by Islam *et al* where the absence of vaccinations was associated with an anthrax outbreak [16]. Routine vaccination policy is one of the better strategies for prevention and control of anthrax [17]. After reporting of the first animal death by the villagers, the veterinary extension officer performed a post post-mortem, made a diagnosis of Theileriosis and people were assured that the meat was safe for consumption. This misdiagnosis was probably due to poor knowledge on anthrax signs as there were no previous animal cases in the area, consistent with a study by Swai *et al* 2010 [18]

We found out that more cases had gastrointestinal symptoms as compared to the cutaneous form. This is contrary to the studies that have shown that the cutaneous form is the most common form accounting for 95% of cases globally [5, 19]. After notification of the first case, active

case finding was commenced. During this active search, health care workers had a high index of suspicion, and this might explain the higher number of gastrointestinal cases reported. These gastrointestinal cases were defined by their history epidemiological linkage. Although the gastrointestinal form is usually associated with fatalities, the symptoms range from sub clinical, mild, moderate to severe [20]. The index case whose diagnosis was initially missed had moderate symptoms however he was symptomatically well managed by aggressive fluid resuscitation. During this current outbreak the early identification and treatment of the other GI cases might help explain the good patient comes, consistent with a study by Nakanwagi *et al* 2020 [21]. However, it is also possible that more than one pathogen has been involved in this outbreak.

The study came up with four statistically significant risk factors for contracting anthrax inclusive of, eating under cooked meat, belonging to a household with cattle deaths, assisting with skinning anthrax infected carcasses and preparing infected meat. Like the findings of several studies with regards to religion, belonging to a religion with restrictive beliefs was protective, however this was not statistically significant [22]. These restrictive religions did not allow consumption of meat from animals that had died from unknown causes. Therefore, members of such religion were less likely to consume the infected meat. The position of most anthrax lesions was on the fingers and the face. The reasons are that the hands especially the fingers are mainly in contact with the infected products while the face is exposed during consumption of probably under cooked meat.

The small sample was a limitation, and this could have reduced the some of the measures of association. We did this study during the outbreak, and this affected the knowledge of the respondents as the health proportion activities were being carried out at the same time. Our other limitation was that of inability to perform culture for our stool samples and molecular testing.

Low vaccination coverage of livestock and an initial low index of suspicion contributed to the outbreak and to the ongoing risk in the community. The timely response by both the human health and veterinary department contributed to the excellent patient outcomes. Eating under-cooked meat, skinning without personal protection, and having a history of dead cattle in the homestead were found to be risk factors for contracting anthrax.

We recommend commencement of routine annual cattle vaccinations against anthrax by the Veterinary department. There is need for establishment of a zoonotic committee that meets regularly outside the outbreak periods for prevention of future epidemics. We also recommend that both the human health and veterinary departments conduct on job trainings on anthrax for their workers. The health promotion department should continue with health education on anthrax in the whole district.

## Acknowledgments

We would like to acknowledge the following for making this study a success:

Residents of Tengwe, Hurungwe District

The Provincial Medical Directorate, Mashonaland West Province

The District Medical Officer, Hurungwe District

The District Veterinary Officer, Hurungwe District

Community Nursing Sister, Hurungwe District

## Author Contributions

**Conceptualization:** Theresa Hamutyinei Dhliwayo, Prosper Chonzi, Collen Madembo, Tsitsi Patience Juru, Addmore Chadambuka, Notion Tafara Gombe, Gerald Shambira, Chukwuma David Umeokonkwo, Mufuta Tshimanga.

**Formal analysis:** Theresa Hamutyinei Dhliwayo, Prosper Chonzi, Collen Madembo, Tsitsi Patience Juru, Addmore Chadambuka, Notion Tafara Gombe, Silvester Chikerema, Gerald Shambira, Chukwuma David Umeokonkwo, Mufuta Tshimanga.

**Investigation:** Theresa Hamutyinei Dhliwayo, Addmore Chadambuka, Silvester Chikerema, Gerald Shambira, Mufuta Tshimanga.

**Methodology:** Tsitsi Patience Juru, Notion Tafara Gombe, Chukwuma David Umeokonkwo.

**Resources:** Prosper Chonzi.

**Supervision:** Prosper Chonzi, Collen Madembo, Tsitsi Patience Juru, Addmore Chadambuka, Notion Tafara Gombe, Silvester Chikerema, Gerald Shambira, Chukwuma David Umeokonkwo, Mufuta Tshimanga.

**Writing – original draft:** Theresa Hamutyinei Dhliwayo, Prosper Chonzi, Tsitsi Patience Juru, Notion Tafara Gombe, Silvester Chikerema, Gerald Shambira.

**Writing – review & editing:** Theresa Hamutyinei Dhliwayo, Collen Madembo, Tsitsi Patience Juru, Addmore Chadambuka, Notion Tafara Gombe, Silvester Chikerema, Gerald Shambira, Chukwuma David Umeokonkwo, Mufuta Tshimanga.

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
