## [Decision Letter · Decision Letter 0]

28 Sep 2022

PONE-D-22-23756Anthrax outbreak investigation in Tengwe, Mashonaland West Province, Zimbabwe, 2022PLOS ONE

Dear Dr. Chadambuka,

Thank you for submitting your manuscript to PLOS ONE. After careful consideration, we feel that it has merit but does not fully meet PLOS ONE’s publication criteria as it currently stands. Therefore, we invite you to submit a revised version of the manuscript that addresses the points raised during the review process.

Based on the reviewers comments, I suggest to make the necessary amendments for further review.

We look forward to receiving your revised manuscript.

Kind regards,

Debdutta Bhattacharya

Academic Editor

PLOS ONE

Reviewers' comments:

Reviewer's Responses to Questions

**Comments to the Author**

1. Is the manuscript technically sound, and do the data support the conclusions?

Reviewer #1: Yes

Reviewer #2: Partly

2. Has the statistical analysis been performed appropriately and rigorously? 

Reviewer #1: Yes

Reviewer #2: Yes

3. Have the authors made all data underlying the findings in their manuscript fully available?

Reviewer #1: Yes

Reviewer #2: No

4. Is the manuscript presented in an intelligible fashion and written in standard English?

Reviewer #1: Yes

Reviewer #2: No

5. Review Comments to the Author

Reviewer #1: Overall the investigation is being carried out by following the One Health approach and it is the need of time to address the zoonotic diseases with a joint approach from the health, veterinary and forest/environment departments. The authors have mentioned that they have initiated the response to outbreak investigation within 24 hours of getting the information about the outbreak.

However there are minor changes to be made in the current form of manuscript and the changes are listed below -

Line 223: Descriptive Epidemiology section is left blank. Please add the information about DE of outbreak investigation.

Line 245-246: Please rewrite the sentence "cases had more knowledge as compared to cases". Meaning not clear.

Line No 291: Remove extra word "January"

Overall there is no information with regards to the human sample collection and their laboratory diagnosis for confirmation of 36 human anthrax cases. Please add this information to support the statement you have written in result and discussion.

The authors have collected the samples of human cases or only information? If so then how do they get to know that who all were the positive for anthrax and where they were being tested. These information is very important for readers and to support the investigation.

What specimen was collection from human cases and what test was being performed for confirmatory. Plz add this inform as well

Line 284-285: Please elaborate the use of One Health approach and formation of investigation team (No of members included from each dept.). You may refer Bhattacharya et. al 2021 for more relevance on One Health Approach for human anthrax.

Reviewer #2: Comments:

Correct the misspelt and italicize Bacillus anthracis (or B. anthracis) throughout the manuscript.

The writing language is very poor and inconsistent. It should be rectified thoroughly before acceptance.

Introduction

Line 96-98: Did the authors consider any other departments like the forest or environment sector while implementing the One Health approach? If not, why? Is there any role of wild animal interaction in the Tengwe area? If so, do the authors have any data about wild animal cases during that time?

Line 115-116: Confusing sentence. Rephrase it

Materials and Methods

Line 143 – The genus must be in uppercase.

Inclusion criteria must be based upon case definition, not only on the consenting of the study participant.

What type of biochemical test was performed? Provide the details. What about Gram staining? Why not RT-PCR or conventional PCR?

Results

Line 225: Why not all 36 cases were interviewed? Were they not willing to participate or what?

Line 245-246: “For the other symptoms…..” correct the sentence.

Line 247-257 –Only symptomatic treatment of patients and vaccination of animals do not contribute One Health approach. What prevention measures were taken to stop the spread of infection in the village environment? It is not clear how the authors implemented the One Health approach.

Line 253-254: “veterinary department for confirmation of anthrax…….” If so, what was the justification for that many cattle deaths in a short span given by the veterinary department? In discussion, the authors stated that the disease in the first dead cattle was declared as Theileriosis by the vet department. Did they state the same for all 17 cattle death?

Line 254: Not “reagents”, change it to “methods”

No result was given about the laboratory test results of the collected samples. How the authors confirmed that the outbreak was due to B. anthracis. Did they collect samples from each human case to confirm the presence of the pathogen? Or just based on the symptoms and case history, they simply predicted the outbreak as anthrax?

Socio-demographic characteristics of controls are missing in Table 1.

The authors didn’t mention the source of the outbreak in humans. I hope the authors have questioned that during the outbreak investigation. Was there any incidence of dead animal consumption among the residents as GI anthrax appeared to be pre-dominant?

Line 268: As the risk of contracting anthrax was higher in females due to maximum involvement in cutting the dead livestock, is that their job responsivities found in that region? Moreover, getting GI anthrax (as found in a higher ratio in this investigation) is primarily related to the consumption of uncooked meat, not to the preparation. Is it so that they also consumed more than the male population?

Discussion

Line 296: change “from” to “form”

Line 297: “Upon notification of the first case………” very confusing sentence making. Make it clear.

Line 311: “…..therefore members of the church were protected from getting…” what do the authors want to mean?

Line 315: There are many more limitations like culture facility, molecular testing etc. to be mentioned.

6. PLOS authors have the option to publish the peer review history of their article (what does this mean?). If published, this will include your full peer review and any attached files.

Reviewer #1: **Yes: **HARI RAM CHOUDHARY

Reviewer #2: **Yes: **Debaprasad Parai

---

## [Author Response · Author response to Decision Letter 0]

31 Oct 2022

PLOS One

Dear Editor(s)

Please find below our responses to comments by reviewers. Thank you for taking time to review. The effort is greatly appreciated

Response to comments from reviewers

Reviewer #1: 

Overall, the investigation is being carried out by following the One Health approach and it is the need of time to address the zoonotic diseases with a joint approach from the health, veterinary and forest/environment departments. The authors have mentioned that they have initiated the response to outbreak investigation within 24 hours of getting the information about the outbreak.

However there are minor changes to be made in the current form of manuscript and the changes are listed below -

Comment 1: Line 223: Descriptive Epidemiology section is left blank. Please add the information about DE of outbreak investigation.

Response 1: We have added a sentence under this section.

Comment 2: Line 245-246: Please rewrite the sentence "cases had more knowledge as compared to cases". Meaning not clear.

Response 2: We have rephrased the sentence to ‘More cases compared to controls were able to identify the other anthrax symptoms. 

Comment 3: Line No 291: Remove extra word "January"

Response 3: We have removed the term January disease and left the term Theileriosis.

Comment 4: Overall there is no information with regards to the human sample collection and their laboratory diagnosis for confirmation of 36 human anthrax cases. Please add this information to support the statement you have written in result and discussion.

Response 4: Human stool specimens were taken from people who had presented with gastrointestinal symptoms, however none was cultured for laboratory confirmation due to unavailability of reagents at the laboratory.

Comment 5: The authors have collected the samples of human cases or only information? If so, then how do they get to know that who all were the positive for anthrax and where they were being tested. This information is very important for readers and to support the investigation.

Response 5: The authors collected the stool samples, but laboratory analysis was not done. These GI cases were defined by their epidemiological linkage. For the cutaneous cases the case definition for the country does not require laboratory analysis.

Comment 6: What specimen was collection from human cases and what test was being performed for confirmatory. Please add this inform as well

Response 6: Stool was collected but analysis was not done due to inability of the laboratory to perform culture. Therefore, case definition for those who also had cutaneous symptoms was through clinical diagnosis as per the country’s guidelines. Those who only had diarrhoeal symptoms were defined by their epidemiological linkage.

Comment 7: Line 284-285: Please elaborate the use of One Health approach and formation of investigation team (No of members included from each dept.). You may refer Bhattacharya et. al 2021 for more relevance on One Health Approach for human anthrax.

Response 7: The investigating team which visited the villages consisted of 4 members from the Human health, 3 members from the veterinary department and one member from the agriculture department. At district level the zoonotic committee comprised 11 members from the human health, 8 from the veterinary department and 3 from the agriculture department.

Reviewer #2: 

Comment 1: Correct the misspelt and italicize Bacillus anthracis (or B. anthracis) throughout the manuscript.

Response 1: We have made these corrections and they are highlighted in the revised manuscript.

Comment 2: The writing language is very poor and inconsistent. It should be rectified thoroughly before acceptance.

Response 2: We have gone through the whole document to rephrase most of the sentences

Comment 3: 

Line 96-98: Did the authors consider any other departments like the forest or environment sector while implementing the One Health approach? If not, why? Is there any role of wild animal interaction in the Tengwe area? If so, do the authors have any data about wild animal cases during that time?

Response 3: Yes, the environmental sector was part of the team that went into the field. In this area there was a report of a buffalo herd that had passed through this area prior to the outbreak. However, there were no reported cases of anthrax from the game reserve.

Comment 4: Line 115-116: Confusing sentence. Rephrase it

Response 4: These farms have been downgraded to small scale and for the past four years, no routine annual vaccinations by the Veterinary department were being done. 

Comment 5: Line 143 – The genus must be in uppercase.

Response 5: This has been corrected.

Comment 6: Inclusion criteria must be based upon case definition, not only on the consenting of the study participant.

Response 6: Noted, thank you. I have revised the inclusion criteria to this “People of all age groups who were residents/visitors of Tengwe between 16 December 2021 to 13 February 2022 who developed clinically compatible symptoms of anthrax, were epidemiologically linked to a confirmed environmental exposure and consented to be part of the study.”

Comment 7: What type of biochemical test was performed? Provide the details. What about Gram staining? Why not RT-PCR or conventional PCR?

Response 7:

Noted, thank you. The procedure was as follows:

Soil samples were collected from carcass burial sites. Dried meat samples were collected from the door-to-door inspections. Isolation of B anthracis from soil and dried meat was done according to the method of Fasanella et al. (2013) with some modification where 7.5g of soil/ dried meat was mixed with 22.5ml sterile deionized water and 0.5% Tween 20. After vortexing, the supernatant was incubated at 65°C for 20mins. 1ml/plate of the supernatant was incubated in B anthracis selective (PLET) media at 37°C for 36-48hrs. Colonies that appeared small, white, round, and domed were subcultured in sheep blood agar for overnight incubation. Grey-white irregular colonies that were Gram positive, non- haemolytic, catalase positive, oxidase negative, non-motile and had a positive Mcfadyean reaction were regarded as B anthracis.

At the time of the study, due to resource constraints we could not perform PCR.

Comment 8: Line 225: Why not all 36 cases were interviewed? Were they not willing to participate or what?

Response 8: Only one person was not willing to participate. The other 4 were not available at their homesteads after two consecutive visits.

Comment 9: Line 245-246: “For the other symptoms….” correct the sentence.

Response 9: Noted thank you. The sentence now reads as ‘More cases compared to controls were able to identify the other symptoms of anthrax. ‘

Comment 10: Line 247-257 – Only symptomatic treatment of patients and vaccination of animals do not contribute One Health approach. What prevention measures were taken to stop the spread of infection in the village environment? It is not clear how the authors implemented the One Health approach.

Response 10: Only one carcass of suspected infected animals was disposed under the supervision of a veterinary officer. Soil samples were taken by the Veterinary department for analysis. A quarantine order for closure of all abattoirs and butcheries was issued on the 10th of January 2022 by the veterinary department. A search of the suspected infected animal products was done in homes where some dried meat was found and disposed by burying with the help of area veterinary officers and environmental health workers. Soda of chloride was not available for disinfection of carcasses and infected soil. Health education sessions were done with all the three departments available. Communication was done with the wild parks department who reported no anthrax cases during this period. 

Comment 11: Line 253-254: “veterinary department for confirmation of anthrax…….” If so, what was the justification for that many cattle deaths in a short span given by the veterinary department? In discussion, the authors stated that the disease in the first dead cattle was declared as Theileriosis by the vet department. Did they state the same for all 17 cattle death?

Response 11: They did not state the same for all the 17 cattle. After the initial misdiagnosis of Theileriosis, the community was assured that the meat was from these dying cattle was safe to eat and no further reporting was being done to the veterinary department. Upon presentation of the first human case and with the institution of the one health response that is when retrospective information on the total number of cattle that had died was collected. 

Comment 12: Line 254: Not “reagents”, change it to “methods”

Response 12: We have changed this.

Comment 13: No result was given about the laboratory test results of the collected samples. How the authors confirmed that the outbreak was due to B. anthracis. Did they collect samples from each human case to confirm the presence of the pathogen? Or just based on the symptoms and case history, they simply predicted the outbreak as anthrax?

Response 13: The diagnosis was based on the clinical presentation of the cases who also had cutaneous symptoms. The guidelines for the country do not recommend laboratory confirmation for cutaneous anthrax. For those that had gastrointestinal symptoms only, the diagnosis was based on case history and epidemiological link to a confirmed environmental exposure.

Comment 14: Socio-demographic characteristics of controls are missing in Table 1.

Response 14: Noted thank you. We have added them

Comment 15: The authors didn’t mention the source of the outbreak in humans. I hope the authors have questioned that during the outbreak investigation. Was there any incidence of dead animal consumption among the residents as GI anthrax appeared to be pre-dominant?

Response 15: Noted thank you. The authors did question this. Yes, there was an incidence of dead animal consumption. Thirty cases (97%) had a history of dead animal consumption. This was more so because the residents were reassured initially that the meat from these dead animals was safe to eat.

Comment 16: Line 268: As the risk of contracting anthrax was higher in females due to maximum involvement in cutting the dead livestock, is that their job responsivities found in that region? Moreover, getting GI anthrax (as found in a higher ratio in this investigation) is primarily related to the consumption of uncooked meat, not to the preparation. Is it so that they also consumed more than the male population?

Response 16: Noted thank you. The variable was cutting and cooking infected meat. The cutting by these women is that of cutting the meat into smaller pieces for cooking. It is the women ‘s responsibility to do this.

Comment 17: Line 296: change “from” to “form”

Response 17: Noted thank you. We have done so.

Comment 18: Line 297: “Upon notification of the first case………” very confusing sentence making. Make it clear.

Response 18: Noted thank you. I have rephrased the sentence to “After notification of the first case, active case finding was commenced. During this active search, health care workers had a high index of suspicion, and this might explain the higher number of gastrointestinal cases reported”

Comment 19: Line 311: “…. therefore, members of the church were protected from getting…” what do the authors want to mean?

Response 19: Belonging to a religion with restrictive beliefs was protective, however this was not statistically significant. These restrictive religions did not allow consumption of meat from animals that had died from unknown causes. Therefore, members of such religion were less likely to consume the infected meat.

Comment 20: Line 315: There are many more limitations like culture facility, molecular testing etc. to be mentioned.

Response 20: Noted thank you. We have included them.

---

## [Editor Report · Decision Letter 1]

18 Nov 2022

Anthrax outbreak investigation in Tengwe, Mashonaland West Province, Zimbabwe, 2022

PONE-D-22-23756R1

Dear Dr. Chadambuka,

We’re pleased to inform you that your manuscript has been judged scientifically suitable for publication and will be formally accepted for publication once it meets all outstanding technical requirements.

Kind regards,

Debdutta Bhattacharya

Academic Editor

PLOS ONE
---

## [Editor Report · Acceptance letter]

1 Dec 2022

PONE-D-22-23756R1 

Anthrax outbreak investigation in Tengwe, Mashonaland West Province, Zimbabwe, 2022 

Dear Dr. Chadambuka:

I'm pleased to inform you that your manuscript has been deemed suitable for publication in PLOS ONE. Congratulations! Your manuscript is now with our production department. 

Kind regards, 

on behalf of

Dr. Debdutta Bhattacharya 

Academic Editor

PLOS ONE